# Functional Aspects of Hypothalamic Asymmetry

**DOI:** 10.3390/brainsci10060389

**Published:** 2020-06-19

**Authors:** David Sandor Kiss, Istvan Toth, Gergely Jocsak, Zoltan Barany, Tibor Bartha, Laszlo V. Frenyo, Tamas L. Horvath, Attila Zsarnovszky

**Affiliations:** 1Department of Physiology and Biochemistry, University of Veterinary Medicine, 1078 Budapest, Hungary; toth.istvan@univet.hu (I.T.); jocsak.gergely@univet.hu (G.J.); barany.zoltan.balazs@univet.hu (Z.B.); bartha.tibor@univet.hu (T.B.); frenyo.laszlo@univet.hu (L.V.F.); 2Department of Animal Physiology and Animal Health, Szent Istvan University, Faculty of Agricultural and Environmental Sciences, 2100 Gödöllő, Hungary; tamas.horvath@yale.edu (T.L.H.); zsarnovszky.attila@mkk.szie.hu (A.Z.); 3Division of Comparative Medicine, Yale University School of Medicine, New Haven, CT 06520, USA

**Keywords:** neuroendocrine lateralization, melanocortin system, estrous cycle, circadian rhythm

## Abstract

Anatomically, the brain is a symmetric structure. However, growing evidence suggests that certain higher brain functions are regulated by only one of the otherwise duplicated (and symmetric) brain halves. Hemispheric specialization correlates with phylogeny supporting intellectual evolution by providing an ergonomic way of brain processing. The more complex the task, the higher are the benefits of the functional lateralization (all higher functions show some degree of lateralized task sharing). Functional asymmetry has been broadly studied in several brain areas with mirrored halves, such as the telencephalon, hippocampus, etc. Despite its paired structure, the hypothalamus has been generally considered as a functionally unpaired unit, nonetheless the regulation of a vast number of strongly interrelated homeostatic processes are attributed to this relatively small brain region. In this review, we collected all available knowledge supporting the hypothesis that a functional lateralization of the hypothalamus exists. We collected and discussed findings from previous studies that have demonstrated lateralized hypothalamic control of the reproductive functions and energy expenditure. Also, sporadic data claims the existence of a partial functional asymmetry in the regulation of the circadian rhythm, body temperature and circulatory functions. This hitherto neglected data highlights the likely high-level ergonomics provided by such functional asymmetry.

## 1. Introduction

The mammalian brain is an organ with symmetrical organization. Anatomically, the mid-sagittal plane cuts the brain into two identical halves with a mirror-symmetric organization in the halves’ histology. However, careful comparison of the two sides shows a number of macro- and microanatomical differences between the sides, which include asymmetrical densities and distribution of distinct types of neurons, the size of specialized circuits or regions, etc.

Morphological differences imply accompanied functional asymmetries. Paul Broca, a French physician was the first to report lateralized functioning in the brain [1]. Broca demonstrated that aphasia (impairment of language, affecting the ability to talk, comprehension of speech, and the proficiency to read or write) is caused by injuries in the left frontal gyrus (exclusively). Follow-up studies have revealed that the two telencephalic hemispheres are devoted to the regulation of distinct biological functions, thus each of them is dominant in different aspects. Now, it is widely accepted that this sort of functional lateralization of the cerebrum ensures the optimal integration of different cognitive processes, such as fine motoric movements of hands, spatial relation, perception and processing of sensory inputs, speech, etc.

The lateralized task-management of the hemispheres provides an elegant solution for “ergonomic use” of brain resources. As an evolutionary tendency to optimize energy utilization, it was reasonable to assume that other brain areas also display functional asymmetry. Indeed, after the initial findings of cortical lateralization, studies have described functional asymmetries in other brain parts such as the hippocampus, habenulae and thalamus, not to mention the characteristic localization of spinal tracts and associated gray matter regions [2,3,4].

Similar to other brain areas, functional asymmetry of the hypothalamus has also been demonstrated as early as the late 1970s [5,6]. Interestingly, these early findings were followed by only very few investigations. Most of these studies described certain aspects of the hypothalamic control of reproduction that showed functional lateralization. Those findings suggest that either the hypothalamic regulation of reproductive functions alone, or as physiologically linked functions, combined with one or more other homeostatic functions are governed from distinct halves of the hypothalamus. Even now, available data on this topic is very scattered and mostly appears as only incidental data or a byproduct of studies focusing on a different aspect.

Functional aspects of the hypothalamic asymmetry could be at least in part coupled with micro-morphologic lateralization. Such morphological basis of the functional asymmetry might manifest in the synaptic plasticity (synaptic remodeling) of the hypothalamic neuronal circuitries previously described as a crucial event in the hypothalamic regulation of the ovarian cycle [7,8,9]. Unfortunately, at the time of our afore cited studies the idea of a possible asymmetry in the hypothalamic synaptology had not yet been born. A rapid fluctuation in the synaptic rearrangements is also inherent to neuronal circuits that govern energy expenditure and food intake (reviewed in [10]). Again, the latter study was reported before the idea of the possible morphological hypothalamic lateralization and, therefore, it has been not documented whether these synaptic events occur with a unilateral dominance. Conclusively, in the lack of such targeted synaptological assessments, further experiments are necessary and would be crucial to determine which functions are regulated through hypothalamic synaptic remodeling.

Asymmetry of the neuroendocrine system has been last reviewed in 1997 [11]; however, several new studies have been reported since then to support the idea of at least partially lateralized neuroendocrine regulation. Here, we attempt to collect and summarize all the notable information available on the hypothalamic asymmetry. 

## 2. Hypothalamic Functions

The hypothalamus is part of the basal forebrain, belongs to the diencephalon and is connected to the pituitary gland by the pituitary stalk. The two hypothalamic sides contain like-named nuclei showing a (mirror) symmetric tissue architecture. The corresponding nuclei of the two sides are thought to (collectively) integrate the same peripheral and central signals that strengthen the idea that the hypothalamus (i.e., the two sides of it) works in unity as a diencephalic functional module.

The hypothalamus is involved in the mediation of endocrine, autonomic and behavioral functions, such as reproduction [12,13,14], thermoregulation [15], thirst [16], hunger [17,18,19,20], memory [21,22,23], circadian rhythm [24,25], stress-responses [26,27], and heart rate [28]. The hypothalamic regulation of different physiological processes is based on distinct hypothalamic morphological structures and “biochemical kits” (sets of molecules used in the regulation of distinct functions). On the other hand, however, there is a well-known overlap in the aforementioned regulatory circuits: there are neuron subpopulations that participate in the regulation of more than one physiological functions (also see below). For example, the arcuate nucleus is a hypothalamic center for the regulation of reproductive functions, but also for the initiation of food intake, etc. Further, estrogen, besides coordinating female reproduction, also plays a key role in the regulation of appetite and energy expenditure as an anorexigenic factor because there are estrogen responsive cells in both neuron populations [29,30,31].

Based on the above-mentioned phenomenon, hypothalamic processes are strongly interrelated and interdependent; however, for didactic reasons, we are going to discuss them separately.

## 3. Reproductive Regulation

One of the most important, at the same time most investigated roles of the hypothalamus is the regulation of reproductive functions. Intrahypothalamic pathways controlling the ovarian cycle (through the regulation of gonadotrophin-releasing hormone (GnRH) and consequential pituitary LH release) converge on GnRH containing neurons in the left and right hypothalamic sides to maintain the normal—also cyclic in females —reproductive activity. However, hypothalamic functions, being physiologically strongly linked, are regulated by neuronal populations whose partial morphological overlap (neurons participating in more than one regulatory function). This attribute of the hypothalamic machinery should be considered during the below discussion of the previously reported functional aspects of hypothalamic asymmetry.

### 3.1. Females

Among the pioneers, Gerendai et al. examined the asymmetry in female reproductive functions by hemi-gonadectomy and found vastly higher GnRH concentrations in the right side of the hypothalamus compared to the left side (in female rats) [5]. Based on these results, they proposed that the central asymmetry is due to a direct, unilateral neural connection between the brain and peripheral organs. In line with these findings, experimental manipulation of the right side of the hypothalamic-pituitary-gonadal (HPG) system proved to be significantly more efficient than that of the left [32,33,34,35]. These phenomena suggest that non-central aspects should also be considered for a better and much more complex understanding of the brain asymmetry.

Among these early morpho-functional experiments, impact of hypothalamic asymmetry on gender-specific differentiation and behavior was also studied. For example, it turned out that left- and right-sided estradiol implants have different effects in neonatal female pups [36], and unilateral estradiol injection evokes more intense hypothalamic responses if applied on the right side causing more pronounced lordosis behavior [37,38]. Seeking for the mechanism leading to the asymmetrical functioning, von Ziegler and Lichtensteiger found that aromatase activity in neonatal female rats is higher on the left side of the hypothalamus [39] (see Figure 1A). Later, McCormick and Singh described that the control of sexual responsiveness might also be influenced by the asymmetric distribution of progestin receptors between the hypothalamic hemispheres [40], and based on their results, the authors suggested that hypothalamic asymmetry might be linked to the cortical asymmetry of progestin receptors described earlier. They were the first to use the term “hypothalamic hemispheres” instead of “hypothalamic sides,” suggesting that there is an interdependent division of functions between the left and right hypothalamic sides, similarly to that observed in the cerebral cortex.

As experimental methods have improved, more reports show that the direction and degree of hypothalamic lateralization may be reflected by the consecutive phases of the estrous cycle and that the timing of ovulation might also be determined by lateralized actions of the neuroendocrine hypothalamus. In 1994, Moran et al. showed decreased (or even ceased) ovulation rate in female animals whose right anterior hypothalamic area (responsible for GnRH-release) was surgically lesioned [41]. In the second half of the 1990s, Lopez et al. designed an experiment placing pilocarpine implants into the preoptic and anterior hypothalamic areas to mimic unilateral acetylcholine effect on ovulation [42]. To briefly summarize their results, on the day of estrous, ovulation was blocked if pilocarpine implants were placed into the left side; on days 1 and 2 of diestrus, implants in both sides were effective. In proestrus, ovulation was blocked only by implants placed into the right side of the hypothalamus. The same group recently suggested that tonic-phasic GnRH-release and the secretion of gonadotropins are normally regulated by the right preoptic and anterior hypothalamic areas [43,44].

Recently, a new aspect of hypothalamic asymmetry research emerged, when our research group measured mitochondrial metabolism separately in the left and right hypothalamic sides [8,9,45]. An aspect of metabolic asymmetry in the hypothalamus was already shown by using glucose utilization technique. They described a higher metabolic activity on the right hypothalamic side without further explanation of the functional implications [6]. Taking these early results into consideration, our research group developed a method to measure mitochondrial metabolism from relatively small brain areas, such as hypothalamic sub-regions [46]. Using this method, intact and later ovariectomized female rats were examined in order to determine the effects of normal cyclic reproductive events on the degree of the hypothalamic metabolic status [8,9]. We described new results that indicated a complex mechanism by which the left and right hypothalamic sides are able to regulate different homeostatic and reproductive processes in an asymmetric manner. In intact, normal cycling rats, we found an estrous phase-dependent metabolic lateralization that shows a right-sided dominance. Estrogen substitution of ovariectomized rats (regardless of satiety states of the animal) caused remarkably higher proportion of sided animals, in which the left and right sides’ capability to react to estrogen seem to significantly differ, with a right-sided dominance (see Figure 1A).

Synapses connecting distinct hypothalamic areas rearrange during a remarkably rapid remodeling driven by endocrine and paracrine effects of both local neurohormones and peripheral signals such as the gonadal estrogen feedback. Due the high energy needs of these micro-morphologic changes, synaptic events are reflected in the fluctuation of the intensity of cellular metabolism and dynamic changes in mitochondrial activity. Indeed, our results describing the estrogen-related hypothalamic asymmetry can be nicely appended to the previously described estrogen-dependent mechanisms (estrogen-induced synaptic plasticity, estrogen-induced alterations in mitochondrial structure and function such us ATP synthesis [7,47,48]. These suggest that research methods may need to be reconsidered so as to get a clearer picture of side-linked (“hemispheric”) synaptic and mitochondrial functions within the hypothalamus.

Taken together the afore-mentioned results, we found strongly estrogen-dependent asymmetric metabolic changes in the hypothalamus. These changes might well be the measurable metabolic parameters of the simultaneously occurring process that has been termed as “estrogen-induced synaptic plasticity” (EISP) and events that lead to the short-term neuronal plasticity-related to GnRH peak [7,49].

### 3.2. Males

The regulation of male reproductive processes are based on the very same morphological basis (HPG axis and GnRH neurons), but the main difference is that in males, due to the early masculinization of the brain, the hypothalamus is not able to respond to rapidly growing estrogen levels with a surge in GnRH and consequently, luteinizing hormone (LH) levels; thus, gonadal steroids have only negative feedback effects. Due to this developmental difference, the main difference compared to females is the lack of reproductive cyclicity.

Early experiments in the 1970s already indicated an asymmetry in the neuroendocrine regulation of reproductive functions in male rats. An asymmetric distribution of GnRH-immunoreactive cells was detected in male mice with a right-sided dominance [5]. Interestingly, five years later, another study found the same asymmetric distribution of GnRH cells [50]. Soon, Bakalkin et al. [51] verified the original observation of Gerendai et al. [5]. Their results also revealed that GnRH content was higher in the right hypothalamic side only if samples were taken in the morning (beginning of the resting period of the animals) (see Figure 1A).

As in the case of females, hemigonadectomy (unilateral castration) helped in understanding of the hypothalamic asymmetry in male animals, and unilateral connection has also been found between the gonads and the hypothalamic sides [52,53]. These studies also suggest an age-dependent lateralization in the central regulation, and in steroid response of the two testes [54].

It was a further addition to our knowledge on hypothalamic asymmetry that exposure to a mild cold stress (two hours before sacrifice) elicits asymmetric response in the GnRH release (increased on the right side, while the left side remained unaffected) [51]. This finding suggests that the hypothalamic cholinergic system, a key player in stress and reproductive behavior also functions in an asymmetric way, just like in female animals [55].

Finally, on the contrary to female rats, mitochondrial activity measurements failed to demonstrate any reproduction-related metabolic asymmetry in male hypothalami [8].

## 4. Regulation of Food Intake and Energy Metabolism

The main neuronal populations regulating food intake and energy homeostasis are also located in the hypothalamus. Based on their mechanism of action, these neurons can be divided into two groups—orexigenic (increasing hunger and food intake) and anorexigenic (decreasing hunger and food intake). The orexigenic and anorexigenic neurons in both hypothalamic sides form complex circuitries, known as the melanocortin system [18].

There are reports to suggest that unilateral neural pathways connect feeding centers to hypothalamic structures with asymmetric functions [56,57,58]. It is also well known that estrogen is not only a sex hormone, but also plays roles in the regulation of many (mainly trophic) functions, including the regulation of energy homeostasis (foodintake, energy expenditure, etc.). These well-known facts together with recent evidence that estrogen evokes the metabolic activation in the hypothalamus in an asymmetric fashion suggests that the hypothalamic regulation of food intake may be just as lateralized as the regulation of reproductive functions [8,9,45].

The first evidence for hypothalamic asymmetry with regard to control of food intake appeared in 1996 [59]. This study showed that unilateral electrostimulation of the lateral hypothalamic area results in altered feeding behavior depending on which of the sides were stimulated: stimulation of the left hypothalamus had a lower threshold and higher stability of food-seeking motivation in rabbits. Later, this left-sided dominance in food-seeking behavior have been confirmed, when electrical activity of the left and right hypothalamic sides of cats was measured during food-related operant reflexes [56]. In this study, the authors described that the asymmetrical neuronal activity of hypothalamus fully correlates with neuronal activity of the frontal cortex (a part of the reward-system of the brain) that suggests unilateral neuronal pathways connecting the hypothalamic hunger-centers to the cortical regions. Later, similar unilateral connections have also been described between the hypothalamus and other parts of the reward-system [57,60]. This data, even if not direct proof for an asymmetric functioning, strongly implicates a lateralized task management in food intake-related hypothalamic functions, and also suggests that the monitoring of other, functionally linked brain areas may lead to a better understanding of hypothalamic lateralization.

In consonance with the above data, our study on fed and fasted (24 h food-deprivation) female rats showed asymmetric metabolic changes between the two hypothalamic sides [9]. In females, due to the overwhelming effect of the presence or absence of gonadal steroids, the exact roles of the left and right sides in the regulation of food intake and energy-balance could not be fully established, therefore, we repeated the experiments on male animals. In these experiments, fasting resulted in left-sided metabolic dominance, while in case of ad libitum feeding, the right side showed higher mitochondrial metabolism, and the degree of asymmetry gradually followed the state of satiety [45]. This implies that the well-studied dynamic interactions between orexigenic and anorexigenic hypothalamic centers differ on the left and right sides. Although it seems that most probably, orexigenic and anorexigenic neuronal activations occur in both hypothalamic sides, results suggest that the satiety state (anorexigenic activation) of the animal primarily depends on the right hypothalamic side, whilst orexigenic activation of the melanocortin system dominates on the left hypothalamic side (see Figure 1B).

As mentioned earlier, related to the reproductive function, hypothalamic circuits involved in regulation of energy expenditure and food intake remain plastic through the adulthood (reviewed in [10]). Critical components of the melanocortin system rapidly respond to nutrient availability and peripheral hormones, such as leptin, ghrelin and gonadal steroids that evoke altered firing activity and synaptic remodeling in proopiomelanocortin and neuropeptide+agouti-related protein producing neurons. Nevertheless, to date, there is no data whether this sort of responsiveness of the melanocortin circuits shows any lateralization.

## 5. Regulation of Circadian Rhythm

The suprachiasmatic nucleus (SCN), the master clock of the body, is a paired hypothalamic nucleus located bilaterally on the dorsal side of the optic chiasm. Neurons projecting from the nucleus control the endogenous fluctuations that occur in the body with a rhythmic pattern which begins a cycle in about every 24 h following the environmental light–dark (day–night) periods [61].

As expected, the regulation of the circadian rhythm also shows some degree of left-right asymmetry. Most research on the regulation of the circadian rhythm was carried out on golden hamsters kept in constant illumination resulting in a phenomenon termed as “splitting”. Several species of mammals and birds, if exposed to constant lighting, can exhibit a disrupted activity of the circadian pacemakers leading to a behavioral arrhythmicity. This arrhythmicity is characterized by a dissociated (or “split”) locomotor or physiological activity in which a single daily period of activity is divided (or “split”) into two periods of equivalent activities (around 12 h apart) with a resting period between them [62,63]. Data reveal that the splitting phenomenon is attributed to the existence of the two separate but equivalent left and right SCN oscillators working oppositely (coupled in antiphase, i.e., shifted about 12 h from each other) [64,65,66,67]. Investigating the antiphase oscillation in split hamsters, de la Iglesia et al. found asymmetric expression levels of clock genes: *Per* gene was unilaterally expressed on either the left or the right side, while *Bmal1* was only found on the contralateral side to *Per* [68]. The same laterality was found with regard to the *arginine vasopressin* and *c-Fos* genes, as well. Later, Mahoney et al. evaluated the contribution of neural pathways to the determination of the circadian oscillation peripherally through assessing expression of the above clock genes [69]. Results verified the asymmetric expression of clock genes in the SCN; however, a moderate asymmetry was detected in the peripheral organs as well, indicating that, besides neural pathways, other mechanisms may also influence peripheral oscillators.

Locomotor activity and GnRH release share a common circadian pacemaker in the suprachiasmatic nucleus (SCN), thus de la Iglesia et al. speculated that in split hamsters, the activity of the left and right GnRH neurons may reflect the sequential activity of neurons in the ipsilateral SCN [70]. Indeed, they found that behaviorally split female hamsters exhibit a left–right asymmetry in the activity of GnRH cells, and that this sided GnRH action is bound to the asymmetric and consecutively alternating activity of the split left and right SCN. On the other hand, unilateral atropine injection in rats could not confirm the existence of the ipsilateral neural connections [71,72]. The aforementioned idea of de la Iglesia et al. may lead to the speculation that functional asymmetry in one anatomical structure leads to similar functional asymmetry in a morphologically and functionally linked other structure. Although we are not aware of any existing evidence supporting this phenomenon, it is definitely worth keeping in mind the possibility of such a scenario as duplicate anatomical structures seem to exist for a reason other than to exert the exact same functions, and relevant reports on functional asymmetry all suggest that such functional lateralization must be supported by an adequately organized anatomical buildup (i.e., appropriate neuroanatomical connectivity pattern). Thus, theoretical conclusions in this matter may serve as the compass of future structural-functional investigations to prove or confute the existence of such machineries in the CNS.

At the same time, there is evidence to suggest that both hypothalamic sides maintain separate circadian oscillators and the SCN functions independently on the left and right sides [73]. Further electrophysiological studies in rats also confirmed phase differences between the left and right SCNs that may affect the circadian timing system [74].

When examining the hypothalamic regulation of food intake, our research group also revealed some insights to the asymmetric nature of circadian events. In ad libitum-fed animals, the left hypothalamic side showed a higher metabolic activity around the day-to-night and night-to-day transitions (see Figure 1C). While these findings support the aforementioned idea of the de la Iglesia’s group, our experiments demonstrating direct functional asymmetry in the hypothalamus both during food-uptake under scheduled light-dark program and a timed feeding program independent of the aforementioned illumination program further strengthens that morphologically and functionally interconnected brain centers entrain each other into temporally synchronized morpho-functional units [45].

## 6. Hypothalamus–Pituitary–Thyroid Axis and Thermoregulation

Besides the above functions, the hypothalamus also governs autonomic responses to temperature changes [75,76,77]. During thermogenesis, parvocellular neurons in the paraventricular nuclei release the thyrotropin-releasing hormone (TRH), a mediator of both metabolism and feeding modulation [78]. TRH modulates the activity of the thyroid gland as well as the uncoupling proteins in brown adipose tissue and muscles through pituitary thyroid stimulating hormone [79]. Beyond TRH-mediated thermogenesis, the dorsomedial hypothalamus (DMH) also sends neural inputs to the rostral raphe region that contains sympathetic premotor neurons providing the principal excitatory input to spinal neurons controlling the activity of thermoeffectors [80].

There is a well-established connection to the regulation of food intake and energy balance, thus one could easily presuppose that the TRH and thermoregulation is just as lateralized as the other physiological processes detailed above. Despite of this, only one study has been published that directly examined the hypothalamus from this aspect [81]. In this study, the authors indicated asymmetrical hypothalamic TRH production in humans, with higher concentrations measured on the left side (see Figure 1D).

Further research on the thyroid gland demonstrated obvious side-linked differences regarding the size, vascularization and innervation of the gland [82,83]. Furthermore, Lewiński et al. found that ceasing the connection between the hypothalamus and the left side of the thyroid gland (left-sided de-afferentation) decreases proliferative activity (basal mitotic index) in thyroid folliculli of both lobes [84]. These observations (asymmetric TRH concentrations in the hypothalamus, direct autonomous innervation to the thyroid lobes, asymmetric consequences of autonomic denervation of the thyroid lobes manifested in basal mitotic index) suggest that a functional asymmetry also exists in the regulation and function of the hypothalamus-thyroid system.

## 7. Regulation of Circulatory Functions

The hypothalamus can also influence heart rate, vasoconstriction, and stress reactions through neural pathways projecting to the lateral medulla [85]. Cardiovascular responses evoked by different stressors are mediated by the DMH. Relevant pathways descending from the DMH target the rostral ventrolateral medulla and the raphe pallidus, transmitting the sympathetic vasomotor commands and the sympathetic cardiac commands, respectively [86].

Based on anatomic lateralization in the pathways from DMH, functional lateralization in the control of cardiac functions was studied by Xavier et al. [87]. They found asymmetrical responses to unilateral DMH injections, indicating that the cardiovascular functions are also lateralized, and dominantly, the right side of the hypothalamus controls changes in heart rate during emotional stress. Using the same method, they also demonstrated increased cardiac contractility (inotropic effect) and other cardiovascular parameters after right DMH injection, while the left-sided injections were less potent [88,89]. In concert with this, stress induced tachycardia could be blunted when the right (but not the left) DMH was blocked by nano-injected antagonist (kynurenic acid) (see Figure 1E).

## 8. Immunological Functions

The hypothalamus plays a central role in the induction of immune responses via neural and humoral pathways to either enhance or depress the immune response [90,91,92].

First, Betancur noted an asymmetry in the control of the immune system when investigated how a lesion to the cortical area of the brain affects natural killer cell activity [93]. The natural killer cell activity dropped markedly after a lesion to the left side; in contrast, right-sided ablation had no effect on activity. In support of this finding, Delrue et al. found hypothalamic asymmetry in the brain monoamine metabolism induced by systemic injections of lipopolysaccharide, a stressor leading to suppression of the immune response [94]. The result showed increased serotonin metabolism in the hypothalamus and hippocampus, but only on the left side. Interestingly, they also registered increased serotonin turnover in the medial hypothalamus observed in right-handed and ambidextrous mice but not in left-handed animals (see Figure 1F). They concluded that an immune challenge could induce responses similar to that caused by stress, and that these responses are asymmetrically expressed along the cortex-hypothalamus axis and appear to be dependent on right and left handedness.

Available data suggests that the side-specific responsiveness depends on the actual state of the immune response. While the left side may be involved in early-state inflammation, mature immunity may trigger the right side. The lateralization observed in serotonergic metabolism supports the above hypothesis on the left-sided dominance in hyperthermia and related somnolence.

## 9. Conclusions

Despite of its relatively small size, the hypothalamus regulates a multitude of physiological processes in order to maintain the physiological parameters of homeostasis and reproduction. The hypothalamic regulation of most of these relevant neuroendocrine processes have been well studied and understood. However, the question of how the multitude of regulatory afferents are entrained into and assigned to the hypothalamic nuclei to precisely orchestrate the simultaneous regulatory functions in their known complexity has received little, if any, attention. Until recently, the hypothalamus was believed to be an unpaired brain structure, in which the two symmetrical sides equally control the same functions. Therefore, this scientific “dogma” meant that the several roles (functions) to regulate homeostatic processes are (also equally) crammed into each half of the hypothalamus to work simultaneously, in tandem.

At this point, it seems that the hypothalamic sides are able to regulate the same functions but with different intensity (functional asymmetry). This gives us a handy solution for the evolutionary question: should the hypothalamus work in a duplex manner or could it spare at least some of its resources. In physiological circumstances, if the two sides are intact and fully functioning, the hypothalamus, by a yet unclear mechanism, is able to outsource functions to only one side. On the other hand, in case of a unilateral lesion, the uninjured side can further regulate the given homeostatic process. The clear understanding of this phenomenon, in other words, the better understanding of hypothalamic functions together with an established future in vivo diagnostic imaging method of the hypothalamus (e.g., real-time thermal monitoring, high resolution PET, etc., please see below) carries the potential to help in improving therapies used in hypothalamus-linked health conditions.

As described above, the topic of hypothalamic asymmetry has not been studied as intensely as it would have been necessary, however, by now it has become a rapidly developing research area and represents a very interesting field of neuroscience. Based on the limited number of papers, it appears that many researchers, after obtaining valuable and interesting data indicating asymmetric functioning, cease to carry on with their research in this field. This may be due to the enormous complexity of the hypothalamic functions, by which this extremely small brain area is able to orchestrate the normal vegetative and reproductive processes of the organism. Broad and systematic morpho-functional analyses could clarify the nature and significance of the functional lateralization of this brain area. Recent technologies of simultaneous positron emission tomography and magnetic resonance imaging or computer tomography enable a spatial resolution as low as 1–1.5 mm, which also allows a real time function-based mapping of diencephalic areas on rodent model using relevant radiotracers [95]. This may not be a sufficient resolution in rodent studies of real-time monitoring of hypothalamic subnuclei but may be enough in species with larger bodies, including non-human primates and humans as well. However, in case of laboratory animal studies, in vivo and ex vivo electrophysiological examinations with bilaterally positioned electrodes could also provide a tool to determine whether or not the activation of hypothalamic circuits or neurohormonal release is lateralized [96].

We hope that this review will attract more focus on the topic of the hypothalamic asymmetry. As this special field of research grows, accumulating data indicates that the correct understanding of the hypothalamic regulation of homeostasis requires the full exploration of the nature of task-sharing between the hypothalamic nuclei and the clarification of the involved neural pathways. Studies reporting that disturbances of lateralized functions may take part in the pathogenesis of hypothalamus-linked health conditions (such as infertility, obesity, anorexia nervosa, etc., as it is already indicated in the case of other brain areas [97]) suggest that the determination of the “interhemispheric” connectivity between the left and right hypothalamic sides will be of crucial importance.

## Figures and Tables

**Figure 1 brainsci-10-00389-f001:**
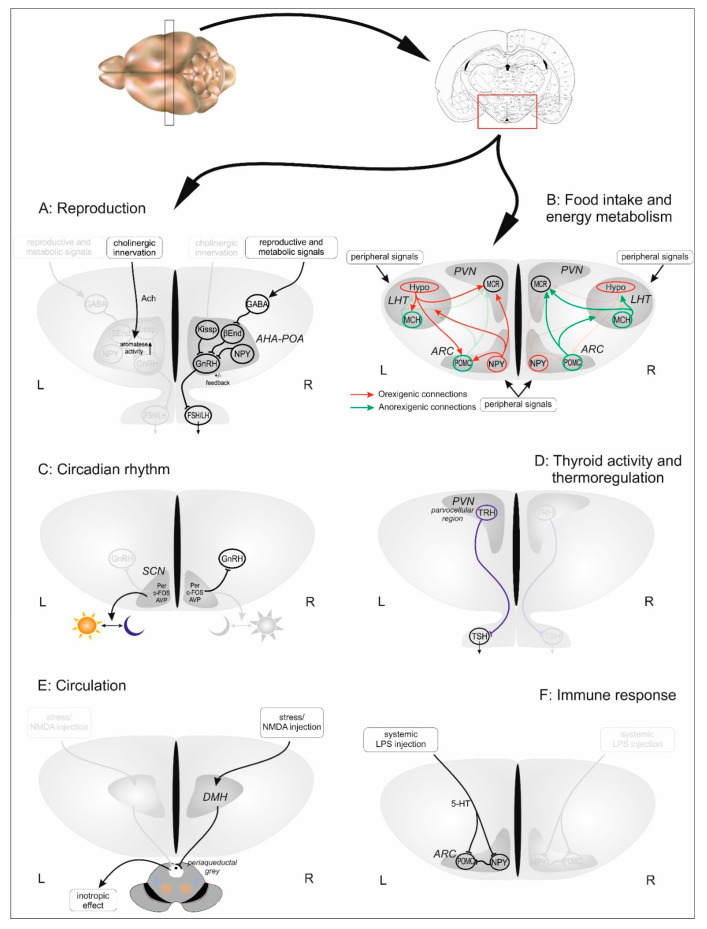
Task sharing of hypothalamic sides as far as it is suggested to date. Figures represent hypothalamic coronal sections, each attributed to a particular function discussed in the text. With regard to a control function those processes are depicted that have been described to be controlled in a lateralized manner. Mechanisms on dominant sides (left (L) or right (R)) are labelled with a brighter color while pale on the contralateral. Abbreviations: 5-HT: serotonin, Ach: Acetylcholine, AHA-POA: anterior hypothalamic-preoptic area, ARC: arcuate nucleus, AVP: arginine vasopressin, βEnd: beta-endorphin, DMH: dorsomedial hypothalamic nucleus, FSH: follicle stimulating hormone, GABA: gamma-aminobutyric acid, GnRH: gonadotropin-releasing hormone, Hypo: hypocretin, Kissp: Kisspeptin, LH: luteinizing hormone, LHT: lateral hypothalamus, LPS: lipopolysaccharide, MCH: melanin-concentrating hormone, MCR: melanocortin receptor, NMDA: N-methyl-D-aspartate, NPY: Neuropeptide Y, POMC: Proopiomelanocortin, PVN: paraventricular nucleus of hypothalamus, SCN: suprachiasmatic nucleus, TRH: thyrotropin-releasing hormone, TSH: thyroid-stimulating hormone.

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
