# Peer review of "Functional Aspects of Hypothalamic Asymmetry"

_brainsci, 2020, doi:10.3390/brainsci10060389_

Round 1
Reviewer 1 Report
This short review summarises historical and modern data focussing on physiological asymmetry of hypothalamic function. The review covers lateral differences of reproductive function, food consumption, energy expenditure, control of circadian rhythms, thermoregulation and circulatory function.
The review is logically structured and well presented. I do not have any critical comments.
Reviewer 2 Report
This is an interesting and thoughtful review that gathers together a wide array of primary reference papers that provide evidence for functional asymmetry in different nuclei/regions of the hypothalamus. The majority of the review focuses on asymmetries involved in reproduction, food intake, energy metabolism and circadian rhythm but the authors do well to include the papers describing lesser-known functional asymmetries – including those that may regulate temperature, circulation and immune response. By necessity (as very little work has been done in these areas), these sections of the review are quite brief, but by pointing them out, the authors raise awareness.
The three major comments I have, that might improve the manuscript are:
- The figure is very useful and specific panels should be referred to throughout the text
- At one point the authors begin to raise the idea that functional asymmetry in one region (the SCN) may govern functional asymmetry in a second (GnRH). I wonder if they could discuss this more clearly – I think the point might be lost on the naïve reader. Clearly, there is no direct evidence for this (the authors point to refs 54-55) – but even so, it helps the reader to think. Ie are the functional asymmetries all mechanistically linked? Or do they arise/operate independently?
- Similar to point 2, Lines 285-290: The discussion of laterality in the thyroid gland threw me a little. It does not fit within framework of rest of manuscript. If kept in, then the authors need to address whether there is laterality in other glands (g pituitary) – and be clear that they are making a rather separate general point , namely that laterality in neuroendocrine systems (including laterality in the hypothalamus) is physiologically important.
Minor comments
- Some of the grammar is imperfect – I suggest key changes below, but in general, the grammar could be tweaked in many places to improve the fluency
Line 14. brain asymmetry is a characteristic of mammalian organisms
Line 21. Despite its paired structure
Line 76-77: like-named nuclei, harbouring hypothalamic neurons, showing…
Line 75: HT – if using this abbreviation
Line 96: roles of the hypothalamus….
Line 97-98: converge on GnRH neurons on the left…
Line 190: The main neuronal populations
- Lines 195-198: Please expand to better-explain why the key findings from papers 49-41 begin to suggest asymmetry in feeding circuits. The terseness with which the key take-home messages of these papers are presented (lines 195-196) stands in contrast to the depth in which the ‘reproductive’ asymmetries were discussed, and this sentence (from 195-198) was hard to follow.
Ref 56: in wrong place. Ref 56 discusses in vitro electrophysiological studies, rather than summarising evidence presented in previous paragraph
Reviewer 3 Report
In this review, Kiss et al. do a nice job summarizing and discussing literature related to the interesting topic: Functional aspects of hypothalamic asymmetry. This review discusses topics that are of great interest in an emerging field, especially given the focus on a critically important brain region (hypothalamus) that is often overlooked by studies examining asymmetrical wiring, signaling and functions. This review will be a great addition and enjoyable read for those studying the hypothalamus; however, the authors do need to address several points that I have listed below. Moreover, to help readers better comprehend the material being discussed and the arguments the authors make, a number of grammatical errors need to be addressed (see below).
- On page 2, line 59 the authors define the hypothalamus as HT, but throughout the manuscript they switch between writing out hypothalamus and using HT. The word hypothalamus doesn’t need to be abbreviated, so delete HT throughout the manuscript and replace it with hypothalamus.
- On page 2, lines 81-83 the authors discuss hypothalamic functions but they don’t include any citations for other reviews that discuss this literature or provide citations for the original studies. This needs to be added. This is particularly important as the authors include “memory” as a documented hypothalamic function and this is not commonly discussed as a well-know/documented hypothalamic function, therefore literature must be cited here to support this statement.
- On page 2, lines 85-89 the authors discuss regulatory circuits with more than one physiological function, but then say “…serving a single regulatory function or are involved in the regulation of more than one regulatory process.” Overall, the argument being made here is confusing and needs to be streamlined.
- On page 3, lines 105-106 the authors define the HPG axis as “…hypothalamus-gonad axis (HPG)…” when this should actually be the hypothalamic-pituitary-gonadal axis (HPG).
- On page 3, line 115 the authors say “…Singh proved…”. Please do not use the word proved. Use “showed” or “described”, etc.
- On page 3, lines 136-137 the authors discuss work from “…our research group…”. Is this peer-reviewed/published research? If so, please include the citation. Otherwise, please describe this work in depth (e.g., methods, results, etc.) and write “data not shown” or include the data in this review.
- On page 6, line 157 the authors discuss “…behaviorally split female hamsters…” and in general the idea of “splitting” and “split” animals need to be explained in further detail.
- On page 6, line 263 the authors again use the word “proved”. Please change this to “showed” or “described”, etc.
- On page 6, line 276 the authors mention the DMH without defining it until later in the manuscript (line 294), so this needs to be fixed.
- On page 7, lines 326-327 under conclusions the authors state: “It is still unknown, how this small-scale brain area is able to orchestrate all these physiological processes”. This language needs to be altered, as the hypothalamus is a well-studied brain region and how it orchestrates many physiological processes have been thoroughly characterized. Therefore, this language should reference how there is a lack of data studying asymmetry in the hypothalamus and how asymmetry plays a role in regulating physiological processes.
- On page 7, lines 338-340 the authors discuss how better understanding hypothalamic asymmetry will help “…offers improved therapies to hypothalamus-linked health conditions”. This is a bit of a stretch, so if the authors want to make this statement they should provide evidence using current literature or discuss this topic in further detail.
- On page 7, lines 347-348 the authors state: “The authors of this resume believe…”. Remove this sentence and discuss how the literature shows/supports/argues for…etc., do not discuss what you “believe”.
- On page 8, line 360 the authors use the word “prove” again. Please change this language.
- The authors do a nice job putting together a figure, however, they do not mention/discuss it once throughout the entire manuscript. The authors should use this figure as a visual aid and refer to it throughout the manuscript when describing findings pertaining to the representations shown in the figure.
- In the figure legend, the authors use the word “proved”. Please change this language.
To help readers better comprehend the material discussed/arguments being made, the following grammatical errors need to be addressed:
- Page 1, lines 14-15: “Founded on genetic and epigenetic effects, brain asymmetry is characteristic for mammalian organism”
- Page 1, lines 23-25: “…the hypothesis on hypothalamic asymmetry…”
- Page 1, lines 25-26: “The clearest findings were collected about the lateralized control…”
- Page 1, lines 26-28: “…functional asymmetry with regard of circadian rhythm…”
- Page 2, lines 49-51: “…also display functional asymmetry to different extent.”
- Page 2, lines 56-59: “…functions are one of the most dominant of hypothalamic processes…”
- Page 2, lines 61-62: “Functional aspects of the hypothalamic asymmetry…”
- Page 2, lines 63-66: “One of the well-described among these is responsible for the control…”
- Page 2, lines 68-69: “Nevertheless, there is no data whether these synaptic events occur…”
- Page 2, lines 71-73: “…the hypothalamic asymmetry.”
- Page 3, lines 96-98: “Pathways controlling the reproductive life converge to the gonadotrophin-releasing hormone (GnRH)…”
- Page 3, lines 119-122: “It is worth to mention that they were the first to use the term…”
- Page 3, lines 123-126: “As experimental methods have become more sophisticated, new data have been raised in the field of hypothalamic asymmetry indicating that the lateralized hypothalamic behavior is also estrous phase-dependent, and the ovulation might also be determined…” 

- Page 3, lines 126-128: “…showed decreased (or even ceased) ovulation rate in female animals with lesion in the right side of anterior hypothalamic area…”

- Page 3, lines 128-129: “…using brain-implantation method to mimic…”
- Page 3, lines 129-132: “…blocked the ovulation; on day 1 and 2 of the diestrus, implants on either side were effective; finally, in proestrus, only implants on the right hypothalamic sides blocked ovulation.” 

- Page 3, lines 137-140: “Some kind of metabolic asymmetry of the brain (including the hypothalamus) was already shown by Glick et al. using 2-deoxy-D-glucose technique describing…”
- Page 3, lines 140-142: “Based on these initial results, our research group, developed…”
- Page 3, lines 144-146: “We described new results indicating a…”
There are a number of additional grammatical errors throughout the remainder of the manuscript (pages 4-8) that I have not mentioned here but that need to be addressed before the manuscript is in a state to be published.
Round 2
Reviewer 3 Report
I appreciate the revisions to the manuscript and in many places the text is now improved. Unfortunately, grammar/use of English language still seems to be an issue and I will leave it to the Editors to decide how best to proceed. In terms of interest and scientific rationale of the paper, I have no concerns.